# Treatment of a Malignant Soft Tissue Tumor Arising in the Vicinity of the Sciatic Nerve with an In-Situ Preparation Technique and Intensive Multidisciplinary Therapy

**DOI:** 10.3390/cancers11040506

**Published:** 2019-04-10

**Authors:** Hisaki Aiba, Katsuhiro Hayashi, Satoshi Yamada, Hideki Okamoto, Hiroaki Kimura, Shinji Miwa, Hiroyuki Inatani, Takanobu Otsuka, Hideki Murakami

**Affiliations:** 1Department of Orthopaedic Surgery, Nagoya City University Graduate School of Medical Sciences, Nagoya 467-8601, Japan; hisakiaiba@yahoo.co.jp (H.A.); ducati888stradasp3-srx600@yahoo.co.jp (S.Y.); yands53okamoto@yahoo.co.jp (H.O.); hiroaki030301@yahoo.co.jp (H.K.); miwapoti@yahoo.co.jp (S.M.); i-hiroyuki@mta.biglobe.ne.jp (H.I.); otsuka-t@tokaigakuen-u.ac.jp (T.O.); hmuraka@med.nagoya-cu.ac.jp (H.M.); 2Department of Orthopaedic Surgery, Kanazawa University Graduate School of Medical Sciences, Kanazawa 920-8641, Japan; 3Department of Orthopaedic Surgery, Fukui-ken Saiseikai Hospital, Fukui 918-8503, Japan; 4Department of Education, Tokai Gakuen University, Nagoya 468-8514, Japan

**Keywords:** soft tissue sarcoma, chemotherapy, hyperthermia, surgery

## Abstract

Preservation of the sciatic nerve is difficult in cases of highly malignant soft tissue tumors closely surrounding the nerve. Herein, we present the first case of preservation of this nerve by combining an in-situ preparation technique (ISP; a technique enabling the preparation of neurovascular bundles without contamination by tumor cells) with intensive concurrent neoadjuvant chemo-radiotherapy with hyperthermia (RHC; radio-hyperthermo-chemotherapy). A 62-year-old man presented with a soft tissue mass in the right thigh and was diagnosed with undifferentiated pleomorphic sarcoma. The tumor arose in the multi-compartment areas of the posterior thigh muscles and was closely intertwined with the sciatic nerve. As preoperative therapy, RHC was performed for surgical down-staging and the tumor partially responded. Afterwards, wide resection of the tumor with preservation of the sciatic nerve using ISP was performed. Following the surgery, there has not been recurrence in the affected site and the functional outcomes of the lower extremity achieved 80% in the Musculoskeletal Tumor Society score. The patient is still alive with disease five years postoperatively. This is the first case in which ISP and RHC procedures were combined for the preservation of the neurovascular structure. Further study is needed for the validation of the feasibility of this method.

## 1. Introduction

Soft tissue tumors arising in the vicinity of neurovascular bundles are major problems for surgeons and sometimes require sacrifice of the neurovascular bundle to allow an adequate margin for possible invasion around the tumor. In-situ preparation (ISP) was first described in 2002 for patients with tumors close to major neurovascular structures [1], and allows the separation of neurovascular bundles and intraoperative evaluation of the surgical margin without contamination by tumor cells around the surgical area. Originally, the ISP technique comprised the following processes: (1) tumor is excised en bloc, including the tumor and neurovascular structures with an adequate wide margin at the proximal and distal ends, and keeping continuity of the neurovascular structures; (2) the resected mass is lifted and isolated from the surgical area by a vinyl sheet (to prevent contamination); (3) the mass is resected through the nearest approach to the neurovascular bundles; (4) the neurovascular bundles are exposed and isolated from the tumor. For the evaluation of the surgical margin, the neurovascular bundles are only permitted to be preserved if they do not completely adhere to the tumor. In this case, the tumor is not considered to be positive for micro-invasion beyond the perineural membrane or vascular sheath; (5) after evaluation, the neurovascular bundles are soaked in pure alcohol for 1–5 minutes, followed by distilled water. The bundles are then returned to the surgical site [1].

To enhance the curability of ISP, we combined it with an intensive neoadjuvant therapy for the preservation of neurovascular bundles; radio-hyperthermo-chemotherapy (RHC), originally introduced at Nagoya City University, was preoperatively performed to eradicate the micro-invasion of the tumor to the neurovascular bundles. We reported favorable outcomes using this technique for local control of the tumor, which enabled preservation of the neurovascular bundles by intentionally reducing the surgical margin [2,3,4]. In this report, we propose the first use of a novel combination technique (ISP + RHC) for the treatment of a highly malignant soft tissue tumor with circumferential invasion to the sciatic nerve.

## 2. Case Presentation

A 62-year-old Japanese man with no past medical history noticed pain and a soft tissue mass on the medial side of the right thigh. Due to gradual worsening of the pain, the patient consulted a local doctor. On physical examination, the patient was not found to have any nerve palsy. On MRI (Figure 1a–e), a soft tissue mass, arising in the multi-compartment areas of the posterior thigh muscles and intertwining with the sciatic nerve, was revealed, and the patient was referred to the Department of Orthopaedic Surgery, Nagoya City University Hospital. A needle biopsy was performed and diagnosed as undifferentiated pleomorphic sarcoma (UPS) by the Division of Pathology of Nagoya City University Hospital (Figure 2). The histological grade was 3 according to the Fédération Nationale des Centres de Lutte Contre le Cancer (FNCLCC) grading system. The patient also underwent computed tomography (CT) and thallium scintigraphy (Figure 1f) for the assessment of the viability of the tumor and detection of metastases; however, there was no evidence of any metastatic lesions. In this study, to assess the accumulation of the tracer, the tumor to background ratio (TBR) is calculated using the formula TBR = (L−B)/B, for accumulation of the tracer at lesion (L) with background (B) [5]. TBR was calculated by experienced radiologists who were independent of this study.

As a preoperative therapy, RHC was proposed for surgical down-staging. The patient was fully informed of the possible adverse events associated with RHC and agreed with our treatment strategy. Before RHC, the artery reservoir was inserted into the superficial femoral artery, and simultaneously, the inferior gluteal artery, partially feeding the tumor, was embolized using a coil. Intra-artery chemotherapy was simultaneously performed with hyperthermia. Concomitant radiotherapy was administered to the primary site for a total of 40 Gy (2 Gy × 20). Details of the RHC procedures are described by Aiba et al. [2], (Figure A1 and Figure A2). Due to grade 2 kidney failure (CTCAE 1.0 criteria), the amount of antitumoral agent was decreased to 80% after three cycles. Furthermore, the white cells count decreased (grade 3) at three cycles, but spontaneously recovered to within the normal range. Anorexia and nausea (grade 2) sometimes occurred during RHC, but no additional fluid therapies or tubal feedings were required.

After five courses of RHC, the uptake of contrast agent on the MRI was attenuated (Figure 3a–c) and the efficacy of chemotherapy was considered to be a partial response (based on modified RECIST criteria [6]). Furthermore, the uptake of the tracer was significantly decreased (Figure 3d). Thus, it was presumed that the micro-invasion of the tumor to the sciatic nerve was attenuated. Also, if the contact between the tumor and the sciatic nerve was not too tight, it would be possible to preserve the sciatic nerve without leaving micro-residue of the tumor using the ISP. For this reason, wide resection using the ISP procedure was proposed for the preservation of the sciatic nerve.

## 3. Surgical Procedure

Skin was incised from the ischial tuberosity to the popliteal fossa. After exposure of subcutaneous tissue, the biceps femoris, semitendinosus, and semimembranosus were dissected at the distal end and turned away from the distal femur. The gluteus major was resected at the level of the lesser trochanter and turned around. The sciatic nerve was found under the resected muscles on both sides and was released within the resected posterior thigh muscles and gluteus major. The nerve and tumor were then elevated and isolated using a surgical drape. In this condition, the tumor was resected gently, and the sciatic nerve was released from the tumor (in this case, the connection of the tumor to the sciatic nerve was stiff and sacrifice of small brunches of the nerve was required). To eradicate the microresidual tumor, the nerve was soaked in alcohol for 5 minutes as part of the ISP technique. The released nerve was then returned to the initial site. An adequate margin (R0) was achieved, except for the sciatic nerve (Figure 4e). Histological evaluation revealed degenerative changes in the tumor with extensive fibrosis and the area of necrosis was about 70% with a negative margin (Figure 4f).

## 4. Postoperative Course

After surgery, the patient underwent five courses of adjuvant chemotherapy with etoposide + ifosfamide + pirarubicin. One and half years after surgery, lung metastases were found and bilateral metastasectomy was performed with endoscopic assistance. Three years after the initial surgery, skip metastasis from the surgical site in the right intrapelvic area around the piriformis and multifocal re-recurrences of the bilateral lung metastases were observed. Pazopanib was prescribed and the patient currently has stable disease. The patient is still alive with disease five years postoperatively. His postoperative function at five years was 80% according to the Musculoskeletal Tumor Society (MSTS) and 79.3% in the Toronto Extremity Salvage Score (TESS). The patient has partial sensory loss in the lower leg (possibly due to the damage during alcohol soaking and sacrifice of the small branches of the nerve) and walks with a short leg brace due to incomplete paralysis of the anterior tibialis and triceps surae.

## 5. Discussion

The function of patients after treatment of a malignant tumor in the pelvis or thigh with en bloc resection of the sciatic nerve is considered to be poor because of severe motor and sensory loss around the leg and foot. This results in intolerable functional deficit and the development of pressure sores. As a result, some clinicians regard resection of the sciatic nerve as an indication for amputation [7]. Meanwhile, others reported that the functional outcomes after preservation of the sciatic nerve were superior to those of patients who underwent amputation [8,9]. Fuchs et al. reported the functional outcomes of 10 patients after resection of the sciatic nerve [10]. The average TESS was 74%. Four patients walked without a cane, while six needed one or two canes. Nevertheless, a comparison of patient satisfaction between amputation and limb-sparing surgery is difficult; therefore, decision-making should be done based on careful discussion and through informed consent.

There are other methods by which the preservation of the sciatic nerve may be attempted. However, the treatment of a tumor in the vicinity of neurovascular bundles is challenging and there are limited choices, including neoadjuvant therapy with intentional marginal resection, or reconstruction with a segmental grafting/artificial nerve. The ISP method was proposed as an alternative to these traditional treatments, which sometimes have a high rate of complications or recurrence due to micro-invasion. This procedure has many advantages in that the neurovascular bundles can be exposed in the isolated area without contamination by the tumor and the resectability of the tumor from the neurovascular bundles can be safely assessed. Generally, although the sheath of vessels and nerves acts as a natural barrier to the tumor, the preoperative determination of invasion beyond these barriers is difficult, even with current modalities, meaning that sacrifice of the vessels and nerves is mandatory in many cases. However, ISP enables direct intraoperative evaluation of invasion beyond the barrier with minimal contamination [1]. Matsumoto et al. have reported resection of a soft tissue tumor with ISP. The recurrence rate was 13.4% [11]. As for the disadvantages of ISP, artery occlusion (11%) and paresthesia (17%) were reported [1,11]. These comorbidities might occur due to the damage to the neurovascular bundles by direct interventions, chemical effects of alcohol, and sacrifice of small nerve branches. Furthermore, the microinvasion to the neurovascular bundles is subjectively evaluated based on the continuity and tightness between the tumor and neurovascular bundles. Thus, in principle, true evaluation of the margin is almost impossible in many cases.

In addition to the procedure, neoadjuvant RHC therapy was added. In this case, the tumor was attached circumferentially to the sciatic nerve with possible involvement beyond the nerve sheath, so it was anticipated that conservation of the sciatic nerve would be difficult without effective neoadjuvant therapy. In our department, the intentional marginal resection is permitted only if the efficacy of RHC is over a partial response in the modified RECIST criteria [3]. In this case, enhancement in Gd+ MRI and the uptake of tracer were markedly decreased. Considering this outcome, preservation of the sciatic nerve was attempted with the combination of ISP. Although there has been no recurrence in the sciatic nerve where the tumor was originally attached, skip metastasis to the proximal area of the sciatic nerve and distant metastases occurred. Thus, further study is needed for the validation of the feasibility of this method. Cumulative experiences of RHC + ISP procedure for difficult cases will provide more concrete evidence and identify the actual roles of these procedures for the preservation of neurovascular bundles.

## 6. Conclusions

We reported a case in which a malignant soft tissue tumor arose in the vicinity of the sciatic nerve. An ISP technique and neoadjuvant RHC were successfully attempted for the preservation of the nerve.

## Figures and Tables

**Figure 1 cancers-11-00506-f001:**
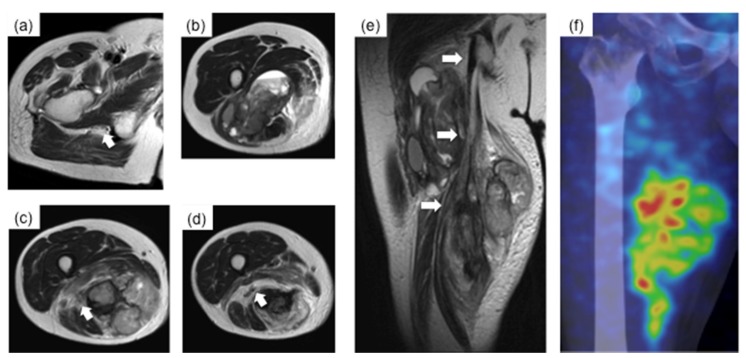
Images before radio-hyperthermo-chemotherapy. (**a**–**e**) (MRI T2 weighted image) revealed that the tumor (12 cm × 10 cm × 31 cm) arose in the posterior thigh muscles (biceps femoris, semimembranosus, and semitendinosus), expanding into the extra-compartment area and surrounding the sciatic nerve (white arrow). (**a**) is at the level of the lesser trochanter, is 5 cm below the lesser trochanter (at this point, the sciatic nerve was almost undetectable), (**c**) is at 10 cm below the lesser trochanter, and (**d**) is at 15 cm below the lesser trochanter. (**e**) is the coronal image of the sciatic nerve. (**f**) (^201^Thallium scintigraphy) showed high uptake of the nuclear tracer in the tumor (mean TBR = 5.3 on early phase).

**Figure 2 cancers-11-00506-f002:**
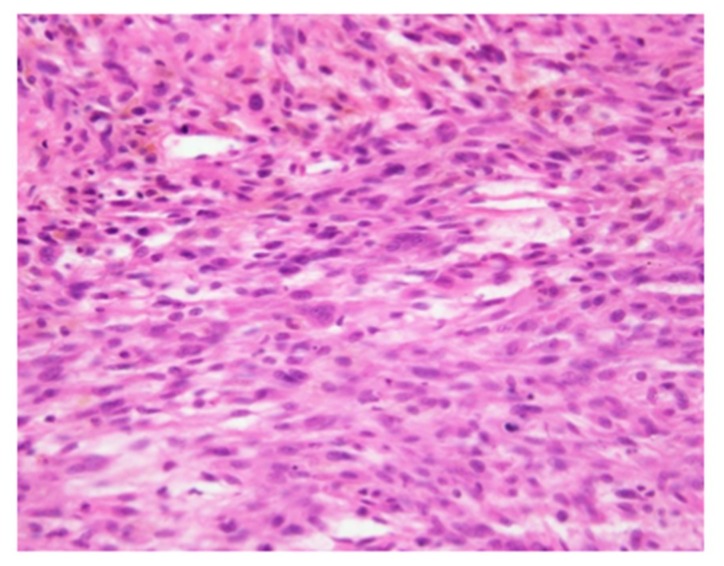
Histological image of the biopsy specimen (H.E. stain, 40×). Spindle cells with atypical nuclei proliferated with giant multinucleated giant cells. The MIB-1 index was 25% and the number of mitotic cells was about 20 cells per 10× area. According to the immunohistochemistry analysis, AE1/3, EMA, S100, myogenin, desmin, D2-40, CD31, and CD34 were negative and the tumor cells were not specifically differentiated. Thus, the patient was diagnosed with undifferentiated pleomorphic sarcoma.

**Figure 3 cancers-11-00506-f003:**
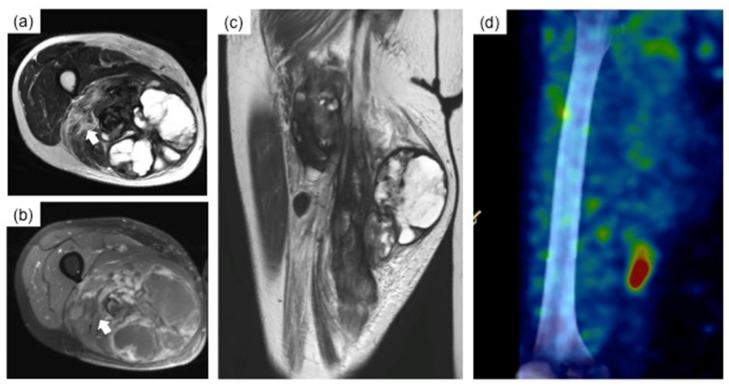
Images after five cycles of radio-hyperthermo-chemotherapy. (**a**–**c**) indicate the cystic and degenerative changes of the tumor. Despite the subtle change in the length of the tumor (12 cm × 10 cm × 25 cm), contrast with gadolinium was only observed in the peripheral area of the cyst, indicating a good response to neoadjuvant therapy. ^201^Thallium scintigraphy revealed attenuation of tracer uptake (mean TBR = 3.2 on early phase, (**d**)).

**Figure 4 cancers-11-00506-f004:**
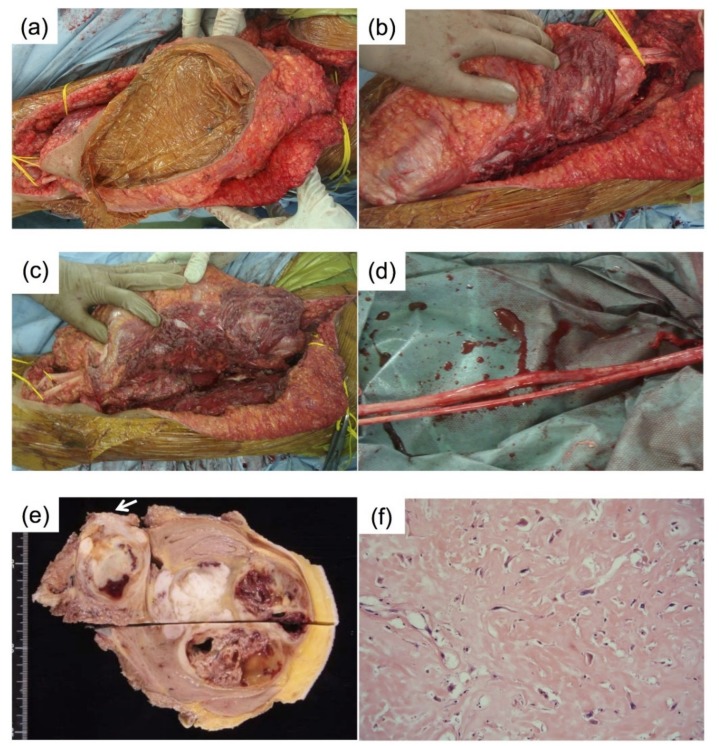
Surgical procedures. Wide resection of the posterior thigh muscles with the tumor was performed (**a**). In the proximal and distal regions, the sciatic nerve was exposed (**b**–**d**). In the area separated by the vinyl sheet, the sciatic nerve was approached from the nearest point to the surface of the tumor and released from the attached tumor. Subsequently, the nerve was treated with alcohol and distilled water. The resected specimen (**e**). The site where the sciatic nerve was originally located (white arrow). Histological image of the resected specimen ((**f**), H.E. stain, 40×).

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
