# Peer review of "Treatment of a Malignant Soft Tissue Tumor Arising in the Vicinity of the Sciatic Nerve with an In-Situ Preparation Technique and Intensive Multidisciplinary Therapy"

_cancers, 2019, doi:10.3390/cancers11040506_

Round 1

Reviewer 1 Report

In this report, the authors present the case of a soft tissue sarcoma patient who presented with a large undifferentiated soft tissue sarcoma of the lower limb.

The oncologic team offered preoperative radio-, hyperthermic-, chemo-therapy, followed by surgery. Surgical resection, which is the focus of this report, was expected to be challenging due to the involvement of the neural bundle, thus posing concerns as to the possibility to preserve limb function.

The “in-situ preparation technique” during the procedure enabled the surgical team to perform limb-preserving tumour resection, followed by favourable functional outcomes. Interestingly, the patient is reported to be alive with disease after 5 years, although both locoregional and distant recurrences have occurred in the meanwhile.

The in situ preparation technique, first introduced in 2002, has already been the topic of extensive publication by the authors of this paper in the course of 2018. In this case report, it was combined with preoperative multimodality treatment in what is claimed to be an original approach.

The paper is well written and rich of technical details, which make it potential interesting. However, I have some remarks:

-  Title: it should be focused on the main topic, possibly mentioning tumor type and tumor site; I would avoid the association with radio-hyperthermic-chemotherapy (in a retrospective report, in fact, it seems to be rather a coincidental combination, rather than a planned strategy).

- Abstract: in my opinion, it should be shortened, but invariably enriched with some technical details (in the present form, it seems not to be informative for the reader).

-  Abstract (and main text): the term “circumferential invasion” is critical and should be clearly and consistently explained.

- Abstract: ISP should be described; moreover, the mention in the text of the different treatments applied should follow the same time sequence with which they were actually applied in the patient.

-  Case report: can you provide pre- and post- CT SUV uptake?

-          Case report (line 101-102): “the uptake of the tracer was significantly decreased (Fig 3d). Thus, it was presumed that the micro-invasion of the tumor to the sciatic nerve was attenuated.” How can you presume this? Rather, microscopic infiltration can be the case despite false reassuring metabolic findings at PET-CT scan. This should be clarified.

-          Case report: what do you mean with “the nerve was excavated”? In a surgical report, I would be unsure of what exactly was performed.

-          Case report: does ISP procedure relies on “nerve excavation” or alcohol soaking?

-          Case report: I feel that a detailed pathological report should be provided in order to establish if the ISP technique had to be applied or not (or better, if there was an indication to its application). By the way, did you perform any intraoperative assessment of critical surgical margins?

-          Case report: the intensity as well as the duration of the toxicities (side effects) mentioned should be reported in greater details and, possibly, also patient-reported feedback.

-          Discussion: pros and cons of the ISP technique should be described and discussed.

-          Discussion: the level of evidence of this intraoperative adjuvant approach cannot be established by this report and the appropriateness of its application could be somewhat questionable outside a clinical study (although it should be recognized that every effort should be pursued to avoid local recurrence in these patients). In order to avoid any over-optimistic statement, I believe that the discussion section should express a critical note on this. Of course, prospective comparative trials are welcomed.          

Author Response

Dear Reviewers,

Thank you for kind remarks and deep insight regarding our manuscript. Based on the suggestion from the reviewers, we have amended our article. We believe these modifications have improved the quality of the article and hope it will now be considered suitable for publication in your esteemed journal.

######################################################

In this report, the authors present the case of a soft tissue sarcoma patient who presented with a large undifferentiated soft tissue sarcoma of the lower limb.

The oncologic team offered preoperative radio-, hyperthermic-, chemo-therapy, followed by surgery. Surgical resection, which is the focus of this report, was expected to be challenging due to the involvement of the neural bundle, thus posing concerns as to the possibility to preserve limb function.

The “in-situ preparation technique” during the procedure enabled the surgical team to perform limb-preserving tumour resection, followed by favourable functional outcomes. Interestingly, the patient is reported to be alive with disease after 5 years, although both locoregional and distant recurrences have occurred in the meanwhile.

The in situ preparation technique, first introduced in 2002, has already been the topic of extensive publication by the authors of this paper in the course of 2018. In this case report, it was combined with preoperative multimodality treatment in what is claimed to be an original approach.

The paper is well written and rich of technical details, which make it potential interesting. However, I have some remarks:

-  Title: it should be focused on the main topic, possibly mentioning tumor type and tumor site; I would avoid the association with radio-hyperthermic-chemotherapy (in a retrospective report, in fact, it seems to be rather a coincidental combination, rather than a planned strategy).

Thank you for the suggestion regarding the title of the article. We changed the title to “Treatment of a malignant soft tissue tumor arising in the vicinity of the sciatic nerve with an in-situ preparation technique and intensive multidisciplinary therapy” (p1, l2)

- Abstract: in my opinion, it should be shortened, but invariably enriched with some technical details (in the present form, it seems not to be informative for the reader).

-  Abstract (and main text): the term “circumferential invasion” is critical and should be clearly and consistently explained.

- Abstract: ISP should be described; moreover, the mention in the text of the different treatments applied should follow the same time sequence with which they were actually applied in the patient.

With consideration of your important remarks, we have shortened the abstract and added technical details (p1, l18-22).

-  Case report: can you provide pre- and post- CT SUV uptake?

-  Case report (line 101-102): “the uptake of the tracer was significantly decreased (Fig 3d). Thus, it was presumed that the micro-invasion of the tumor to the sciatic nerve was attenuated.” How can you presume this? Rather, microscopic infiltration can be the case despite false reassuring metabolic findings at PET-CT scan. This should be clarified.

We are sorry for the confusing description. In this study, instead of the PET-CT, we evaluated the viability of the tumor using thallium scintigraphy. Since the tumor-to-baseline ratio (TBR) is generally used for the assessment of the uptake of tracer, we added statements regarding TBR. (legends of Fig 1 and 3).

- Case report: what do you mean with “the nerve was excavated”? In a surgical report, I would be unsure of what exactly was performed.

We apologize for the confusion around this term. We changed the term “the nerve was excavated” with referral to original article (Fig legend 4, with referring Int J Clin Oncol. 2002, 7(1):51-6.)

- Case report: does ISP procedure relies on “nerve excavation” or alcohol soaking?

We think both are important for the ISP. The former acts to release the tumor and the latter eradicates microinvasion to the nerve.

-Case report: I feel that a detailed pathological report should be provided in order to establish if the ISP technique had to be applied or not (or better, if there was an indication to its application). By the way, did you perform any intraoperative assessment of critical surgical margins?

We added the HE stained image of the postoperative specimen in Fig 4 to improve the clarity (Figure legend 4). With regards to your question about intraoperative assessment of critical surgical margins, for the preservation of the nerve, assessment of microinvasion to the nerve is always insufficient. We described this in the discussion section. (p6, l181-186)

-Case report: the intensity as well as the duration of the toxicities (side effects) mentioned should be reported in greater details and, possibly, also patient-reported feedback.

Thank you for the helpful suggestion. We added information about the side effects of RHC based on the CTCAE criteria.

- Discussion: pros and cons of the ISP technique should be described and discussed. (p6, l181-186).

-Discussion: the level of evidence of this intraoperative adjuvant approach cannot be established by this report and the appropriateness of its application could be somewhat questionable outside a clinical study (although it should be recognized that every effort should be pursued to avoid local recurrence in these patients). In order to avoid any over-optimistic statement, I believe that the discussion section should express a critical note on this. Of course, prospective comparative trials are welcomed.   

Thank you for this suggestion. We added some descriptions in the discussion section. (p6, l187-198)

Reviewer 2 Report

In „Novel surgical technique: In situ preparation 2 technique combined with radio-hyperthermo-chemotherapy for the treatment of a highly malignant soft tissue tumor with circumferential invasion to the sciatic nerve” the authors present a quite interesting case report about a large UPS of the thigh surrounding the sciatic nerve in an elderly patient. The authors preserved the sciatic nerve nerve by combining an in situ preparation technique and neoadjuvant radio-hyperthermo-chemotherapy. The techniques all well described and the margins surrounding the nerve were negative. Unfortunately, the patient developed pulmonary and skip metastases afterwards. However, the functional outcome was very good (assessed through TESS and MSTS) and the patient is still alive with disease after 5 years. It is an very interesting case and the treatment steps seem reasonable. The case report is written well and it deserves publication in a renowned journal.

Some points need to be addressed:

1. Why has the patient a sensory loss in the lower leg. Was the sciatic nerve injured during resection or was the nerve partially resected? Or was it injured by radiation?

2. When the authors assessed the function (TESS, MSTS)? One or 5 years after resection?

3. What was the gradng of the tumor? Please also provide size.

4. How did the treat the skip metastases afterwards? Only pazopanib or did they also resect them?

Author Response

Dear Reviewer,

Thank you for kind remarks and deep insight regarding our manuscript. Based on the suggestion from the reviewers, we have amended our article. We believe these modifications have improved the quality of the article and hope it will now be considered suitable for publication in your esteemed journal.

 In „Novel surgical technique: In situ preparation 2 technique combined with radio-hyperthermo-chemotherapy for the treatment of a highly malignant soft tissue tumor with circumferential invasion to the sciatic nerve” the authors present a quite interesting case report about a large UPS of the thigh surrounding the sciatic nerve in an elderly patient. The authors preserved the sciatic nerve nerve by combining an in situ preparation technique and neoadjuvant radio-hyperthermo-chemotherapy. The techniques all well described and the margins surrounding the nerve were negative. Unfortunately, the patient developed pulmonary and skip metastases afterwards. However, the functional outcome was very good (assessed through TESS and MSTS) and the patient is still alive with disease after 5 years. It is an very interesting case and the treatment steps seem reasonable. The case report is written well and it deserves publication in a renowned journal.

Some points need to be addressed:

1. Why has the patient a sensory loss in the lower leg. Was the sciatic nerve injured during resection or was the nerve partially resected? Or was it injured by radiation?

Thank you for your comments. As you mention, the surgical procedure and radiation were likely reasons for the damage. However, the true causes of the nerve injury were difficult to elucidate because of the multimodal intensive treatment required for local control. We added statements about this in the article. (section 4)

2. When the authors assessed the function (TESS, MSTS)? One or 5 years after resection?

This was done 5 years later but was consistently unchanged 1 year postoperatively.

3. What was the gradng of the tumor? Please also provide size.(section2 and Fig legends 2 and 4))

We added information about it.

4. How did the treat the skip metastases afterwards? Only pazopanib or did they also resect them?

The patient is responding to pazopanib, so for the meantime, we will keep treating the patient with this agent. As you say, some new treatments for soft tissue tumors have become available so we are preparing with assessment of the progress of the tumor. (section 4)

Round 2

Reviewer 1 Report

Dear Authors,

thank you for your work, which has addressed all my requests and improved the manuscript.

The paper is well presented, critical, and rich of useful details.

It will be of interest to the readers.